# How Government Size Expansion Can Affect Green Innovation—An Empirical Analysis of Data on Cross-Country Green Patent Filings

**DOI:** 10.3390/ijerph19127328

**Published:** 2022-06-15

**Authors:** Jun Wen, Lingxiao Li, Xinxin Zhao, Chenyang Jiao, Wenjie Li

**Affiliations:** 1School of Economics and Finance, Xi’an Jiaotong University, Xian 710061, China; wjun19781127@mail.xjtu.edu.cn (J.W.); xinxinzhao@stu.xjtu.edu.cn (X.Z.); 2School of International Economics and Trade, Lanzhou University of Finance and Economics, Lanzhou 730020, China; jiao994422@foxmail.com; 3School of Business Administration, Shandong University of Finance and Economics, Jinan 250014, China; liwenjie@sdcjdx5.wecom.work

**Keywords:** government size, green innovation, two-way fixed effects, inverted U-shaped relationship

## Abstract

The expansion of government size will have dual effects on a country’s green innovation. An appropriately sized government size increases marginal productivity and stimulates the development of green innovation by increasing government expenditure. On the contrary, an excessively sized government creates a huge administrative agency, which not only increases the tax burden but also damages social welfare by excessive intervention. Therefore, the effect of government size on green innovation is not linear. In order to prove this proposition, this study examines the impact of government size on green innovation in 166 countries between 1995 and 2018, using a two-way fixed effects model. The results reveal an inverted U-shaped relationship between government size and the level of green innovation, indicating that optimal government size may maximize a country’s green innovation output. The results further suggest that this inverted U-shaped relationship is mainly influenced by environmental regulations and financial support. Finally, our heterogeneity analysis demonstrates that the inverted U-shaped relationship is more pronounced for countries with high organizational inertia and more R&D expenditure than for those with low organizational inertia and less R&D expenditure. This finding makes up for the research gap between government size and green innovation and provides a reference for countries to formulate the optimal government size to improve the level of green innovation.

## 1. Introduction

Throughout the 20th century, global economic development occurred at the expense of the natural environment. By putting quantity and output value first, the extensive economic growth model of high pollution, high emission, and high consumption ran counter to the sustainable development of society. Decades ago, the Club of Rome in its “Limits to Growth” report warned that “The current economic development model has brought devastating disasters to the world and to humanity itself” [1]. To alleviate the tension between economic development and environmental protection, countries around the world must adopt a green economic development model, characterized by harmony between people and nature.

In the 24th Conference of the Parties to the United Nations Framework Convention on Climate Change in 2018, green technology, green innovation, and implementing the Paris Agreement [2] were put forward as ways to push forward and support the development of a global green economy. In light of this and the ongoing economic shifts from extensive to intensive development, green innovation is at the top of the agenda for all countries.

However, due to its close connection to a multitude of industries, green innovation is highly integrated, flexible, competitive, and risky. Consequently, its development goals may be difficult to achieve if it relies on the market mechanism to maintain the market status quo over the long term. By acting as a “visible hand” to enhance the equilibrium effect of the market mechanism, governments can effectively offset the negative externalities caused by market failures and guide the development of green innovation. Research has shown that factors including government subsidies [3] and legislative constraints [4]. However, no research has touched upon the influence of government size on green innovation.

The term “government size” could be traced back to the French Revolution in 1789. At this time, individuals who supported the king stood on the right side of the parliament while those who supported the revolution stood on the left. The aggregation of the right and that of the left shaped the initial concept of government size [5]. Following the global economic crisis of the 1930s, to address the market failures caused by the overly free market, monetarists (represented by Keynes) emphasized the need for government intervention in the economy and advocated for a larger government and high fiscal deficit to stimulate the economy and maintain prosperity [6]. As a result of economic growth, the political significance of government size faded, while its economic influence became increasingly prominent.

Government size is closely related to a country’s administrative efficiency, economic growth, and social stability. The expansion of government size is based on the principle of economic and social benefits, with the goal of achieving Pareto optimality and optimizing the rational allocation of resources. Furthermore, the expansion of government size is inevitable because of the independence of administrative power, the expansion of social needs, and the expansion of government functions. There are generally three measures of government expansion: the proportion of government consumption in GDP [7], the proportion of government officials in the employed population [8], and the number of administrative agencies [9].

A moderate expansion of government size could better compensate for market failures [10] and guide resources such as funds and talents towards areas in urgent need of development [11], especially in areas with high investment costs and high-risk factors such as green technology innovation [12]. For example, financial support and strengthening of legislative protection could guide more resources to flow to the field of green technology innovation [13,14]. However, the excessive expansion of government size, beyond the critical value, and excessive intervention will inevitably damage social welfare and cause abnormal economic development [15,16], which is similar to the planned economic system.

As the role of government size on a country’s economic development is uncertain, and the impact of government size on green innovation—an indispensable aspect of economic development—is worthy of an in-depth discussion within the academic community. Studies on this topic are vital to support governments, not only to play their role of “providing green technology support and formulating regulations and policies” in green innovation but also to address market failures by forming market-based mechanisms of their own. The latter is particularly important in contexts where the market, or the “invisible hand”, plays a decisive role, while the government, the “visible hand”, leads the way and coordinates with the market. To address this gap, this study draws on cross-country data and empirical analysis to examine the following three research questions. Does government expansion significantly improve a country’s level of green innovation? If so, what are the underlying mechanisms? Will the influence of government size on green innovation vary in response to differences in organizational inertia and the amount of R&D expenditure between countries?

This study makes the following key contributions to the literature. First, this study adopts a different research perspective from previous studies. With regard to government, this study explores the relationship between government size and green innovation by using transnational data—a topic not previously covered in the literature—and is thus both novel and innovative. In addition, while studies on this topic have contended that with the continuous expansion of government size, the level of green innovation in a country presents a nonlinear trend of rising first and then declining, i.e., an inverted U-shaped relationship, we propose that an inverted U-shaped relationship (i.e., the existence of an optimal size of government) is more realistic, in line with objective laws and people’s cognition. Such in-depth research on how government size affects green innovation has both theoretical significance and practical value. Second, the research methods employed in this study expand on those of the literature. Namely, this study uses a two-way fixed effects model to empirically analyze the relationship between government size and green innovation. This model is both scientific and rigorous, enhancing the reliability of the conclusions. In addition, to address endogeneity, the robustness of the benchmark results is tested using Driscoll and Kraay standard errors and systematic GMM estimation. Other methods include the substitution of explained variables, the addition of control variables, the Poisson test, and the negative binomial test. Third, our heterogeneity analysis differs from that of previous studies. Compared with countries with low organizational inertia and less R&D expenditure, the inverted U-shaped relationship is more pronounced for countries with high organizational inertia and more R&D expenditure. This study provides additional evidence for the impact of government size on green innovation and addresses gaps in the literature.

The rest of the study is arranged as follows. The Literature Review section analyzes the theoretical perspectives and formulates our hypothesis. The Data and Methods section introduces the data, variables, and estimation methods, while the subsequent section presents the findings of the empirical analysis. The study concludes with the overall research results and presents policy recommendations.

## 2. Literature Review

### 2.1. Factors That Influence Green Innovation

A country’s green innovation is affected by several factors, which can be summarized by the following three aspects: market, society, and enterprise. From the perspective of the market, the profit that a country’s green innovation makes is determined by market costs [17], market risks [18], market supply [19], and other factors. This means that the green innovation capability of a country improves as the market mechanisms of competition, price, and supply and demand become more developed [20]. With regard to society, factors including the ethics of the social environment and interest groups’ awareness of environmental protection [21] provide positive externalities and create a social environment supportive of green production, a green lifestyle, and green innovation. Such factors may provide a country with a comparative advantage when competing in international markets. When it comes to enterprises, the maturity of the green supply chain [22], financial status, and other factors can motivate enterprises to promote green innovation.

However, some studies have found that factors such as excessive resource investment, information asymmetry, and insufficient R&D expenses will largely inhibit the improvement in green innovation output. For example, redundant resources could provide enterprises with the ability to resist external environmental pressures, but excessive resource input will lead to a waste of resources and will reduce the efficiency of green innovation [23]. The intensification of market information asymmetry hinders the information communication mechanism, thereby reducing the commercialization efficiency of green scientific and technological achievements [24] and inhibiting the output of green innovation. In addition, high R&D expenses and financing constraints are the key economic problems faced by enterprises in green technology innovation [25]. The shortage of funds limits the green innovation behavior of enterprises.

### 2.2. Government and Green Innovation

Research has offered varying opinions on how governments influence green innovation, but these can largely be divided into two opposing views. First, the government can remedy market failures and promote the efficiency of green innovation in a country, and second, government intervention violates the laws of market competition, price, and supply and demand, inhibiting a country’s green innovation output in the long term. Considering this, the following sections elaborate on the impact of government subsidies and environmental legislation on a country’s green innovation.

First is the impact of government subsidies on green innovation. In the market economy, enterprises lack the motivation to work on green innovation due to its externalities. Therefore, to improve green innovation capabilities, government subsidies can counter the insufficient incentives caused by externalities and encourage enterprises to increase their R&D expenditure in green innovation by reducing the costs and risk of doing so [26,27,28]. In other words, government subsidies can encourage enterprises to increase their R&D expenditure in green technology.

However, some research of the manufacturing industry in Europe found that, although government subsidies help to reduce R&D costs for enterprises, they discourage them from investing in R&D on their own, i.e., a crowding-out effect exists [29]. This suggests that governments may fail to have a thorough grasp of corporate innovation, and government subsidies may distort innovation incentives. As such, government involvement may not drive enterprises to increase their R&D expenditure, but rather reduce it, resulting in a crowding-out effect [30].

Second is the impact of the government’s environmental legislation on green innovation. The government of a country can motivate enterprises to protect the environment through the offsetting effect of environmental legislation on innovation. This requires local governments to increase their efforts to improve policies relating to corporate green innovation and alleviate environmental pollution via green technology innovation. Regarding the regulatory effect of the government’s environmental legislation on green innovation, some research pointed out that enterprises prefer to develop and apply clean technologies to control their pollutant emissions and meet higher environmental standards [31]. Similarly, using German panel data, some scholars discovered that administrative capacity, such as laws, regulations, and environmental management, can improve a country’s green innovation output to some extent [32], consistent with the Porter Hypothesis.

However, some studies have shown that the impact of environmental legislation on innovation is not always effective. For instance, some scholars demonstrated that reducing energy costs can bring more benefits to society, but this reduction is unlikely to have much impact on environmental technological innovation [33]. Furthermore, some research also pointed out that strict environmental legislation can promote R&D innovation, but the cost of regulation is far greater than the incremental benefits brought about by such innovation [34,35].

### 2.3. Hypothesis

Government expansion can also increase public service functions, provide society with “positive externality” public services and products (such as transportation, energy, communication, and other types of infrastructure), and effectively make up for the “failure” and “vacancy” of the market [36]. Furthermore, it can have a positive spillover effect by limiting or eliminating monopoly and supporting enterprise development [37]. Therefore, government expansion can help ameliorate both the internal and external environment of green innovation, alleviate the shortage of capital, talent, and environment required for green innovation, and enhance a country’s green innovation.

However, continuous government expansion can lead to a distorted and ineffective allocation of social resources, resulting in the weakening of the positive externalities of public services [38]. Government expansion may also result in government influence in areas that should be characterized by “governance by inaction.” Indeed, the negative impact of such excessive intervention on green innovation has been noted in the literature [39]. In addition, bloated organs, redundant personnel, power rent-seeking, and other problems caused by government overexpansion have increased tax burdens and aggravated corruption [40], resulting in a crowding-out effect harming green innovation [41] and inhibiting a country’s green innovation output. In other words, with continued government expansion, a country’s green innovation capacity will show an inverted U-shaped relationship (i.e., rising first and then falling). Below, we elaborate on the specific mechanisms of environmental regulations and financial support.

Due to limited resources, small governments are unable to develop macro and all-inclusive environmental policies, but they can provide micro and targeted environmental regulations by enabling public power to control pollution and improve the ecological environment [42]. According to the Porter Hypothesis, scientific and reasonable environmental regulations can upgrade green production technology and enhance the degree of green innovation, thus overcoming the limited effect of regulations [43]. However, based on neoclassical economic theory, formalism and bureaucracy increase as the government expands, inevitably leading to more complex and stringent environmental regulations, such as overly high pollutant emission standards and taxes on pollution [44]. In neglecting real-world conditions, such overlapping administrative regulations result in not only a waste of government resources but a surge in the cost of pollution control for enterprises, consequently draining investment in green innovation. Indeed, it is well-known that as the cost impact of environmental regulations rises [45], the output of a country’s green innovation reduces over the long term.

As for financial support, in the initial stages of government expansion, fiscal policies tend to place greater emphasis on green innovation in production and life, and focus on addressing market failures [46]. This can help enterprises overcome capital shortage, financing constraints, and R&D risks, and stimulate green innovation practices. It can also encourage them to take a more active role in green innovation by compensating for the positive externalities caused by green innovation [47]. In addition, financial support can address the fund shortage associated with enterprises’ pollution control expenditure, enabling them to devote more resources to green innovation. However, undue financial support in the form of subsidies, tax exemptions, or fee reductions can make enterprises over-reliant on the government, and thus green innovation becomes unsustainable as the government expands. Once the government cuts investment, enterprises can encounter problems relating to insufficient motivation for innovation, declining output, and low efficiency [48]. Therefore, we propose the following hypothesis: there is an inverted U-shaped relationship between government size and green innovation, all other conditions being equal.

## 3. Data and Methods

Based on panel data from 166 countries between 1995 and 2018, in this study, we empirically analyzed the relationship between government size and green innovation or, more specifically, whether there is a relationship between government consumption, size, and the number of green patents. Data on government size were taken from the Economic Freedom of the World (EFW) database, which has wide-ranging international influence and applicability [49]. The EFW has compiled data on government size and legal systems of property rights, among others, for 166 countries since 1975, all from third-party sources, and has been widely used in empirical research (e.g., the International Country Risk Guide, Doing Business Report of the World Bank) [50,51]. Data on the number of green patents were taken from the OECD Environmental Statistics Database, which includes country-level data on the environment and green technologies. Data for the control variables were obtained from the World Development Indicators (WDI) database.

### 3.1. Explained Variable

Green Patent (Patent): Green innovation R&D, application, and promotion rely on a country’s intellectual property system, in particular, the protection of green patents. The number of green patent applications can be used to measure the achievement of a country’s environmental protection and sustainable development [52]. Some scholars suggested that to distinguish between inter-organizational and intra-organizational relationships caused by non-green innovation, green patents should be regarded as an indicator of green innovation [53]. Therefore, in this study, we drew on the studies of some other scholars [54,55] and used the natural logarithm of the number of environmental green patent applications to measure the level of green innovation across countries.

### 3.2. Explanatory Variables

Government consumption expenditure (Govexpense): Some research discovered that the ratio of government consumption expenditure to GDP can be used to measure the size of a country’s government when studying the relationship between trade openness and government size [56]. Furthermore, countries with higher fiscal expenditures have larger governments, with government consumption expenditure being a major indicator [57]. Inspired by previous studies, in this study, we used the ratio of government consumption expenditure to GDP to measure government size.

Government size (Govsize): According to the definition of EFW, some research proposed that a total index to measure government size can be derived from measures of government transfer, subsidies, investment, taxation, and other sub-indices, as this index not only includes most of the sub-indices related to government size but can also reflect its size in a more comprehensive, complete, and representative way [58]. As such, inspired by previous studies, we adopted the natural logarithm of government size as a second indicator of the size of a country’s government.

### 3.3. Control Variables

GDP per capita (GDP): GDP is the most important statistic in macroeconomics. While GDP growth can solve problems such as financing difficulties, talent shortages, and misallocation of resources for green innovation [59], it can also increase national income per capita and stimulate social groups’ pursuit of a green life when their material needs are satisfied, thus driving the state, society, and enterprises to strengthen their support for green innovation [60]. Therefore, we used the natural logarithm of per capita GDP to measure the level of a country’s economic development.

Population density (Pop): The impact of population and its distribution on green innovation in a country is mainly achieved through two effects: a scale effect and a combined effect. Due to population, the scale effect can increase requirements for the development of production and life, which in turn encourages the country to solve problems such as resource scarcity and environmental degradation via green innovation [61]. In addition, the geographical distribution of a country’s population is important for green innovation, with the combined effect being brought about by talent aggregation [62]. Thus, we used population density to measure a country’s population.

Human capital (HCI): Human capital, specifically innovative talents, is a key factor in the success of green innovation [63]. As the most dynamic element in green innovation, human capital can not only “increase marginal returns and decrease marginal costs” with investment growth but can also provide an inexhaustible impetus for this. Thus, we used the human capital index to measure a country’s human capital.

Exchange rate (Xr): As one of the main external environmental factors for green innovation, exchange rate fluctuations can change relative prices and competition in the international market [64]. This can encourage enterprises to reduce the cost of importing intermediate goods through technological upgrading, changing original production methods, and accelerating new product R&D, in turn affecting the level of technological innovation in a country [65]. As such, we used the natural logarithm of the exchange rate (domestic currency/USD) as the main external factor of green innovation.

Consumer Price Index (CPI): The CPI reflects changes in the price of consumer goods and is usually used to observe a country’s inflation rate. As the CPI rises, the inflation rate will also increase, meaning that in the process of green innovation, the salaries of scientific researchers and the price of raw materials rise accordingly, squeezing green innovation expenditure [66]. We drew on the research of other scholars and selected a country’s CPI (in constant 2017 USD) to measure its inflation rate.

Capital Price (Cp): From design to R&D, and production to application, green innovation relies on a variety of activities requiring substantial investment, such as market development, production, and management innovation. Reasonable capital pricing can offer green innovation a favorable financing environment and strong capital support [67]. Considering this and following the research, we selected the capital price index (in constant 2017 USD) to measure the pricing level of a country’s green innovation market.

Social financing level (Sp): Innovation investment differs from other types of investment in that it is more likely to be affected by financing constraints. The main reasons are the large amount of innovation investment required, its high uncertainty and long payback period, which can result in internal financing failing to meet the needs of green innovation [68]. In addition, information asymmetry is likely to bring about moral hazard and adverse selection, leading to higher financing costs and harming green innovation output. Inspired by the research of some scholars [69], we used stock prices to measure a country’s financing capacity in green innovation.

### 3.4. Descriptive Statistics

The descriptive statistics of the main variables are presented in Figure 1 and Table 1. Table 1 shows that the average value of Patent is 2.808 and the standard deviation is 2.325, indicating a wide variation between different countries in terms of green patents. It is important to clarify which types of countries have more green patents. The standard deviation, minimum, median, and maximum values of Govexp are 2.137, 0.029, 6.000, and 12.957, respectively, showing that government expenditure differs greatly from country to country. The maximum Govsize is 2.245 and the minimum is 0.010, demonstrating that government size varies greatly between countries. In addition, we examined the descriptive statistics of the control variables. Taking GDP as an example, the standard deviation is 14.333, indicating a huge gap in the level of economic development between countries. The maximum Pop is 7.264 and the minimum is 0.019, which is in line with practice, and population density varies greatly across countries.

Table 2 reports the correlations between the main variables and their multicollinearity. As can be seen from Table 2, the variance inflation factor (VIF) of the main variables is less than 5, and Tolerance is greater than 0.2, indicating that the variables are independent and cannot be expressed linearly, i.e., there are no multicollinearity issues [70].

### 3.5. Model Setting

Drawing on the research designs of scholars [71], this study empirically analyzed the relationship between government size and green innovation using the two-way fixed effects approach. This is a useful method, as it can both improve the accuracy of econometric estimates and greatly reduce interference from multicollinearity [72]. The specific model is as follows:(1)$$\;\;\;\;\;\;\;Y_{i,t}=\alpha_{0}+\alpha_{1}Goverment_{i,t}+\alpha_{2}Z_{i,t}+u_{i}+\lambda_{t}+\varepsilon_{i,t}$$
where $\;Y_{i,t}$ is our green innovation variable; $Goverment_{i,t}$ is our explanatory variable of government size; $Z_{i,t}$ a vector of control variables that may affect green innovation; $u_{i}$ indicates time fixed effects; $\lambda_{t}$ indicates country fixed effects; and $\varepsilon_{i,t}$ is the error term.

## 4. Empirical Analysis

### 4.1. Benchmark Regression

The benchmark regression results of the effect of government size on green innovation are shown in Table 3. Table 3 is divided into two groups: columns (1)–(3) use Govexp, and columns (4)–(6) use Govsize as the measure of government size. All regressions control for time fixed effects and individual-specific fixed effects. In columns (1)–(3), after adding the control variables, the regression coefficients of Govexp and Govexp^2^ on Patent are 0.114 and −0.016, respectively, and are significant at the 5% and 1% levels. This indicates that the relationship between government size and green innovation is inverted U-shaped. Taking column (3) as an example, if the other control variables remain unchanged, every 1% expansion of government size will increase the output of green innovation by 11.4%; this value is significant at the 5% level when government size is relatively small. When government size continues to expand, every 1% expansion leads to a 1.6% reduction in the level of green innovation, significant at the 1% level. In columns (4)–(6), after adding the control variables, the coefficients of the primary and quadratic terms of Govsize on Patent are 1.224 and −0.490, respectively, both significant at the 1% level. This indicates that a country’s level of green innovation will first rise and then fall with government expansion, i.e., there is an optimal size of government. In summary, with government expansion, the level of green innovation in a country will show a nonlinear trend, demonstrating an inverted U-shaped relationship. Thus, our hypothesis is confirmed.

### 4.2. Mechanism Test

The empirical results above show that there is an inverted U-shaped relationship between government size and green innovation. This raises the logical question of what causes this inverted U-shaped relationship and, more specifically, what are the mechanisms behind it. Following our theoretical analysis, we propose that environmental regulations, financial support, and other mechanisms contribute to the inverted U-shaped relationship.

#### 4.2.1. Environmental Regulations

The influence of government size, as a core explanatory variable, on environmental regulations may affect the scientific quality of the mediating effect. In consideration of this, we use the environmental regulation index to carry out our regression analysis (Er—Environmental regulation index: Data taken from the OECD database. The higher the absolute value of Er, the stricter the environmental regulation requirements of a country). The results are presented in Table 4. As illustrated in Table 4, after adding the control variables, the coefficients of Govexp and Govsize on Er are 0.803 and 1.900, respectively, and are significant at the 1% and 10% levels. This demonstrates that government expansion greatly improves environmental regulations and promotes green innovation. Moreover, the coefficients of Govexp^2^ and Govsize^2^ are both negative and either highly significant or marginally significant. This indicates that excessive government expansion and too many strict and overly elaborate environmental regulations may waste government resources and cause inefficiency, i.e., there is a negative externality effect. Thus, a deterioration in the green innovation environment experienced by enterprises may lead to a long-term decline in a country’s green innovation output.

#### 4.2.2. Financial Support

In addition to environmental regulations, government size affects the level of green innovation in a country through financial support. To test this, we use fiscal expenditure (Fs) as an indicator of a country’s level of financial support and analyze the impact of government size on green innovation. The regression results are shown in Table 5. After adding the control variables, the coefficients of the primary and quadratic terms of Govexp and Govsize on Fs are 0.053, −0.007, 0.118, and −0.007, respectively, and are either highly significant or marginally significant. This indicates that as governments expand, to enhance green innovation output, enterprises’ green innovation projects are supported in the initial stages by an increase in financial expenditure. However, excessive financial expenditure may result in a culture of dependency among enterprises, discouraging them from independent financing and self-development and, consequently, inhibiting a country’s long-term green innovation.

### 4.3. Heterogeneity Test

The theoretical analysis, empirical test, and mechanism test show that there is an inverted U-shaped relationship between government size and green innovation. In the context of green development, it is crucial to consider in which countries this relationship is more pronounced and clarify how best to determine the optimal size of government to offset the negative externalities of government expansion on green innovation. As such, we examine this inverted U-shaped relationship based on two factors at the national level: organizational inertia and R&D expenditure.

#### 4.3.1. Heterogeneity Results in Countries with Different Levels of Organizational Inertia

Government size refers to the complexity of the internal and external environment of the government, whereby its influence on innovation is essentially the result of the environment. In this regard, the impact of governments of different sizes on innovation differs, just as different organizational characteristics influence innovation in various ways [73]. Compared with countries with low organizational inertia, those with high organizational inertia lead the way in organizational resources and structure, making the inverted U-shaped relationship between government size and green innovation more prominent in these countries.

When the government of a country with high organizational inertia is small, its organization is characterized by great flexibility and high decision-making efficiency. In such a context, there can be smooth communication of environmental regulations and financial support policies between management and technology departments. Both sides can coordinate flexibly, creating an atmosphere that encourages enthusiasm for innovation at the technical level [74]. However, when the government continues to expand, the division of its departmental functions becomes more extensive and mature, leading to high organizational inertia not seen in countries with low organizational inertia. To avoid the sunk and risk costs associated with organizational renewal, governments in such situations place greater emphasis on the predictability and enforceability of the organizational system. The resulting rigidity and other problems can lead to slow or even stagnant organizational change. Moreover, high organizational inertia can contribute to governments being slow to change environmental regulations, financial support, and other policies. As a result, it is likely to cause lock-in and orientation effects on green innovation [75], inhibiting the innovation momentum of enterprises and thus, the rate of green innovation in a country.

In conclusion, different organizational structures have different effects on green innovation. Compared with developing countries, developed nations have higher organizational inertia because they have well-developed organizational structures and cultures. As such, we use dummy variables and divide the sampled countries into developed and developing countries. Developed and developing countries are assigned a value of 1 and 0, respectively. The regression results are presented in Table 6, with Panel A representing the statistics of developed countries and Panel B representing the data of developing countries. In Panel A, when Govexp and Govsize increase, Patent first rises and then falls, which is consistent with the results of our benchmark regression and indicates that the inverted U-shaped relationship between government size and green innovation is prominent in developed countries. In Panel B, although the coefficient of Govsize^2^ is marginally significant, the coefficients of Govexp and Govsize are not significant, indicating that the inverted U-shaped relationship is not obvious in developing countries.

#### 4.3.2. Heterogeneity Results in Countries with Different Levels of R&D Expenditure

A country’s green innovation output is closely related to its R&D expenditure [76]. Compared with countries with low R&D expenditure, the inverted U-shaped relationship between government size and green innovation is likely to be more pronounced in countries with high R&D expenditure, as government size does not affect financial support for green innovation in these countries. As the government invests more in R&D, the scale effect and combined effect of green innovation become more obvious, increasing the country’s green innovation output. However, although government overexpansion can provide green innovation with sufficient funds and talent, it can also trigger the threshold effect between R&D expenditure and green innovation [77]. When this occurs, excessive R&D expenditure can tempt enterprises to invest in more radical and riskier green innovation projects. Such risky behavior not only violates the laws of economic and social development but wastes green innovation resources and, in turn, hampers the improvement of green innovation efficiency.

The level of green innovation in different countries is largely determined by the amount of R&D expenditure. To differentiate the level of R&D expenditure between countries, we divide countries in our sample based on median R&D expenditure: countries with R&D expenditure higher than 7.286 are classified as “high investment countries” (denoted by H_R&D), and those with R&D expenditure less than 7.286 as “low R&D expenditure countries” (denoted by L_R&D). The regression results are presented in Table 7. Panel A reports the results of H_R&D, and Panel B reports those of L_R&D. In Panel A, the primary and quadratic terms of Govexp and Govsize are reversed to the regression coefficients of Patent and are significant at the 1% level. This suggests that the inverted U-shaped relationship between government size and green innovation is significant in countries with high R&D expenditure, consistent with theoretical expectations. As for Panel B, as the government continues to expand, green innovation output does have a nonlinear relationship. In other words, the inverted U-shaped relationship between government size and green innovation is not observed in countries with low R&D expenditure.

### 4.4. Robustness Tests 

#### 4.4.1. Replace the Explained Variable

In addition to the number of environmental green patent applications, the number of applications for water pollution control [78] and climate change [79] can measure a country’s green innovation level. These variables are denoted by Patent1 and Patent2, respectively. The natural logarithm of all patent applications is included in the regression, and the results are presented in Table 8 and are divided into two groups: columns (1)–(4) use Patent1 and columns (5)–(8) use Patent2 as the indicator of green innovation. In columns (1)–(4), after adding the control variables, the coefficients of Govexp and Govsize on the primary and quadratic terms of Patent1 are positive and negative, respectively, and they are highly significant or marginally significant. This indicates that government expansion inevitably leads to a nonlinear trend, or an inverted U-shaped relationship, in green innovation. In columns (5)–(8), like the results in columns (1)–(4), the coefficients of Govexp, Govexp^2^, Govsize, and Govsize^2^ on Patent2 are 0.153, −0.022, 1.267, and −0.543, respectively, and highly significant. This demonstrates that the inverted U-shaped relationship does not change significantly when using a different explained variable. Our benchmark results are therefore robust.

#### 4.4.2. Add New Control Variables

To further verify the robustness of our baseline results, we add new control variables to the original control variables: the proportion of an industry’s added value to GDP (Industry); the proportion of total imports and exports to GDP; and the proportion of scientific researchers to the total population. The results are listed in Table 9. After adding the control variables, the primary and quadratic terms of Govexp and Govsize on Patent are reversed to each other and highly significant or marginally significant. This demonstrates that a country’s level of green innovation exhibits a nonlinear trend. Therefore, the inverted U-shaped relationship does not change substantially following the addition of new control variables. Our benchmark results are therefore robust.

#### 4.4.3. Poisson Test and Negative Binomial Test

Theoretically, the number of environmental green patent applications, as a discrete variable, may not meet the assumption conditions of linear regression models, meaning that the conclusions drawn may not be reliable. As environmental green patents show a discrete distribution, we use the Poisson test and the negative binomial test to further analyze the impact of government size on green innovation. The regression results are shown in Table 10 and are reported separately for each test. We use our original data on the number of environmental green patent applications for both tests. For the Poisson test, the regression coefficients of Govexp, Govexp^2^, Govsize, and Govsize^2^ on Patent are 0.193, −0.015, 6.089, and −1.754, respectively, and are significant at the 5% or 10% level. For the negative binomial test, the regression coefficients of the primary and quadratic terms of Govexp and Govsize on Patent are reversed and highly significant or marginally significant. In other words, a country’s green innovation output shows an inverted U-shaped relationship as the government expands. Our benchmark results are therefore robust.

#### 4.4.4. Systematic GMM Test

Although the two-way fixed effects model based on panel data can alleviate the effect of time-invariant omitted variables, it cannot account for the dynamic effect of the explained variable. Therefore, we adopt a two-step systematic GMM test to solve potential endogeneity [80]. The regression results are shown in Table 11, with columns (1)–(2) using Govexp and columns (3)–(4) using Govsize as the measure of government size. In all columns, after adding the control variables, the p-values of the Sargan test and the Hasen test are greater than 0.1, indicating over-identification. The p-values of the AR (1) test are all less than 0.01, which reveals that there is only first-order autocorrelation in the residual sequence of the sample. Therefore, the two-step systematic GMM test is effective. The results presented in Table 11 further indicate two important findings. First, the coefficient of the first-order lag of Patent is significantly positive at the 1% level, meaning that environmental technological innovation is continuous and dynamic. Second, the coefficients of the primary and quadratic terms of Govexp and Govsize on Patent are reversed to each other and significant at the 1% or 5% level. This is consistent with the results in Table 3. Overall, these results indicate that the inverted U-shaped relationship is more pronounced as government size expands.

#### 4.4.5. Driscoll and Kraay Standard Errors

The Driscoll and Kraay standard errors can be used in fixed effects models to overcome heteroskedasticity in panel data [81]. Therefore, we use the Driscoll and Kraay standard errors to test the robustness of the relationship between government size and green innovation. The regression results are shown in Table 12. As revealed in Table 12, after adding the control variables, the coefficients of the primary and quadratic terms of Govexp and Govsize on Patent are reversed to each other and are highly significant or marginally significant. This means that as a country’s government expands, its green innovation level demonstrates a nonlinear trend. This result is consistent with those of Table 3, indicating that our benchmark results are robust.

## 5. Conclusions and Policy Recommendations

This study examines the impact of government size on a country’s green innovation using unbalanced panel data from 166 countries around the world for the 1995–2018 period. The results reveal that with government expansion, a country’s green innovation output exhibits a nonlinear trend, i.e., an inverted U-shaped relationship. A series of robustness and endogeneity tests show that our baseline results are robust. To determine in which countries this relationship is most pronounced, we use two indicators, organizational inertia and R&D expenditure, and divide the sampled countries accordingly. These two indicators are chosen because they not only reflect government size but also solve the acute problems of scarcity of funds, talent, and markets experienced in green innovation. Surprisingly, the results show that the inverted U-shaped relationship is more pronounced for countries with high organizational inertia and more R&D expenditure than for those with low organizational inertia and less R&D expenditure.

The results of this study are novel. Our research not only finds more factors that affect green innovation from the perspective of governments but also addresses research gaps in the relationship between government size and green innovation using both theoretical and empirical analyses and the perspective of government economics. Moreover, unlike previous linear studies, we find that with government expansion, a country’s level of green innovation has an inverted U-shaped relationship (i.e., there is an optimal government size effect). This finding not only overcomes the shortcomings of previous research and the narrow ideological viewpoints of linear studies, but better reflects the laws of economic and social development. This can, in turn, provide countries with a reference for determining and creating an optimal size of government.

However, restricted by the selection of variable indicators, the two-way fixed effect model selected in this article is relatively simple, which becomes the deficiency of this article and a problem that needs to be improved urgently. In addition, although we examine the inverted U-shaped relationship, we do not include all of the factors that can influence government size, green innovation, and the control variables, such as country’s gross industrial product, education level, etc. This may affect the validity and robustness of our conclusions. Although this research has limitations, it nevertheless provides a useful direction for future studies.

The policy recommendations of this study are as follows. The development of green innovation in a country is inseparable from the shaping of the optimal government size. In particular, it is necessary to optimize the structure of government expenditure, increase investment in public services and infrastructure in the field of green innovation, and create a better investment environment and financial support for green innovation. In addition, in order to prevent excessive intervention of government size on green innovation, the optimal government size needs to build a clean government through budgeting, supervision, and other means; effectively improve the utilization rate of capital, talent, and other resources in the field of green innovation; and prevent resource wasting and damages to social welfare. At the same time, the optimal size of government needs to create a flat, flexible, and highly effective administrative agency to manage the country’s green innovation projects and, as far as possible, similar functions, such as merging business organizations, and avoiding overlapping functions and management confusion caused by problems such as low efficiency and bloated government size in order to better service the country’s green innovation.

Furthermore, for government size—particularly in countries with high organizational inertia and R&D expenditure—there is a need to be cautious about policies that encourage government expansion to reach alarming levels, i.e., the point at which the influence of government size on green innovation turns from positive to negative. The main reason for this is the potential for excessive expansion by the government to eventually lead to even greater negative effects and, consequently, reduce the level of green innovation in the country. In addition, merely relying on government expansion to advance green innovation will not be effective every time. However, it may be possible to reduce the negative effects of government expansion on green innovation by controlling taxation, improving government financing channels, and using other auxiliary measures to build a government with optimal size.

## Figures and Tables

**Figure 1 ijerph-19-07328-f001:**
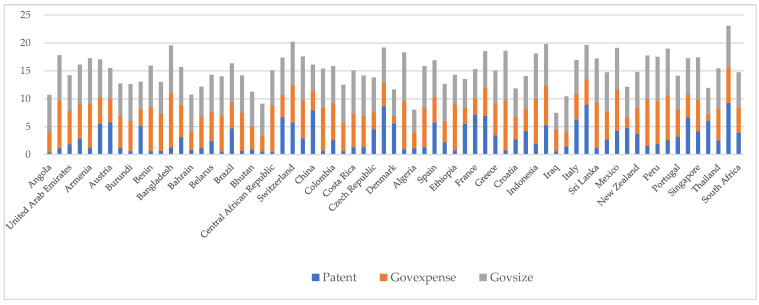
Histogram of government size and green patent applications.

**Table 1 ijerph-19-07328-t001:** Descriptive Statistics of Variables.

Variable	Observations	Mean	S.D.	Min	Median	Max
Patent	2497	2.808	2.325	0.131	2.110	9.791
Govexp	2497	5.809	2.137	0.029	6.000	12.957
Govsize	2497	1.875	0.222	0.010	1.911	2.245
GDP	2497	13.103	14.333	5.331	10.909	16.831
Pop	2497	3.659	4.943	0.019	2.217	7.264
HCI	2497	2.455	0.693	1.049	2.532	4.154
Xr	2497	6.442	8.767	0.001	2.192	12.870
CPI	2497	0.509	0.518	0.084	0.442	23.123
Cp	2497	0.553	0.770	0.006	0.500	33.371
Sp	2497	0.448	0.503	0.065	0.367	20.322

Note: This table provides the descriptive statistics of the main variables, including number of observations, mean, standard deviation, minimum, median, and maximum.

**Table 2 ijerph-19-07328-t002:** Multicollinearity Test.

Variable	Govexp	Govsize	Pop	HCI	Xr	CPI	Cp	Sp	VIF	Tolerance
Govexp	1								2.94	0.339
Govsize	0.651	1							2.28	0.438
GDP	−0.107	−0.069							2.06	0.486
Pop	0.003	−0.055	1						2.07	0.482
HCI	−0.486	−0.184	−0.053	1					1.87	0.534
Xr	0.235	0.189	0.041	−0.196	1				1.07	0.936
CPI	−0.537	−0.306	−0.104	0.626	−0.203	1			4.03	0.248
Cp	−0.316	−0.220	−0.095	0.432	−0.128	0.646	1		2.16	0.463
Sp	−0.441	−0.271	−0.104	0.586	−0.184	0.831	0.748	1	4.74	0.211

Note: This table reports the multicollinearity of the main variables. VIF should be less than 5 and Tolerance should be greater than 0.2, indicating that there are no multicollinearity problems between the variables.

**Table 3 ijerph-19-07328-t003:** Influence of Government Size on Green Innovation.

Variable	Patent	Patent	Patent	Patent	Patent	Patent
	(1)	(2)	(3)	(4)	(5)	(6)
Govexp	0.086 *	0.111 **	0.114 **			
	(1.72)	(2.24)	(2.28)			
Govexp^2^	−0.013 ***	−0.016 ***	−0.016 ***			
	(−3.04)	(−3.86)	(−3.88)			
Govsize				0.939 **	1.216 ***	1.224 ***
				(1.98)	(2.60)	(2.61)
Govsize^2^				−0.345 **	−0.485 ***	−0.490 ***
				(−2.31)	(−3.28)	(−3.30)
GDP		−0.000 **	−0.000 **		−0.000	−0.000
		(−2.13)	(−2.19)		(−0.80)	(−0.77)
Pop		0.008 ***	0.008 ***		0.007 ***	0.007 ***
		(7.64)	(7.59)		(6.45)	(6.32)
HCI		0.396 ***	0.391 ***		0.700 ***	0.684 ***
		(2.90)	(2.84)		(5.00)	(4.84)
Xr		−0.000	−0.000		−0.000	−0.000
		(−0.30)	(−0.35)		(−0.10)	(−0.18)
CPI			−0.094			−0.279 **
			(−0.77)			(−2.20)
Cp			0.032			−0.006
			(0.58)			(−0.09)
Sp			0.107			0.205
			(0.88)			(1.60)
_cons	2.278 ***	0.992 ***	0.984 **	1.700 ***	−0.306	−0.192
	(14.35)	(2.70)	(2.57)	(4.29)	(−0.58)	(−0.36)
Year	Yes	Yes	Yes	Yes	Yes	Yes
Country	Yes	Yes	Yes	Yes	Yes	Yes
N	1820	1726	1720	1936	1797	1791
R^2^	0.416	0.468	0.469	0.384	0.447	0.450

Note: ***, **, and * indicate significance at the 0.01, 0.05, and 0.1 levels, respectively; Values in parentheses indicate t-statistics.

**Table 4 ijerph-19-07328-t004:** Regression Analysis of Government Size on the Environmental Regulation Index.

Variable	Er	Er	Er	Er
	(1)	(2)	(3)	(4)
Govexp	0.772 ***	0.803 ***		
	(4.47)	(4.80)		
Govexp^2^	−0.069 ***	−0.068 ***		
	(−4.81)	(−4.94)		
Govsize			2.379 **	1.900 *
			(2.28)	(1.81)
Govsize^2^			−0.192 **	−0.148 *
			(−2.31)	(−1.76)
_cons	−0.538	2.209	−5.927 *	−1.171
	(−1.15)	(0.87)	(−1.81)	(−0.29)
Control variables	No	Yes	No	Yes
Year	Yes	Yes	Yes	Yes
Country	Yes	Yes	Yes	Yes
N	1444	1444	1445	1445
R^2^	0.310	0.396	0.277	0.359

Note: ***, **, and * indicate significance at the 0.01, 0.05, and 0.1 levels, respectively; Values in parentheses indicate t-statistics.

**Table 5 ijerph-19-07328-t005:** Regression Analysis of Government Size on Financial Support.

Variable	Fs	Fs	Fs	Fs
	(1)	(2)	(3)	(4)
Govexp	0.030	0.053 *		
	(1.14)	(1.88)		
Govexp^2^	−0.006 ***	−0.007 ***		
	(−3.04)	(−3.36)		
Govsize			0.103 **	0.118 **
			(2.22)	(2.39)
Govsize^2^			−0.008 **	−0.007 *
			(−2.08)	(−1.70)
_cons	3.279 ***	3.484 ***	2.684 ***	2.879 ***
	(38.86)	(16.24)	(18.62)	(11.29)
Control variables	No	Yes	No	Yes
Year	Yes	Yes	Yes	Yes
Country	Yes	Yes	Yes	Yes
N	2361	2170	2617	2330
R^2^	0.304	0.316	0.347	0.349

Note: ***, **, and * indicate significance at the 0.01, 0.05, and 0.1 levels, respectively; Values in parentheses indicate t-statistics.

**Table 6 ijerph-19-07328-t006:** Heterogeneity Test: Countries with Different Levels of Organizational Inertia.

Variable	Patent	Patent	Patent	Patent	Patent	Patent	Patent	Patent
	(1)	(2)	(3)	(4)	(5)	(6)	(7)	(8)
Panel A: Developed Countries					Panel B: Developing Countries			
Govexp	0.396 ***	0.435 ***			0.019	0.077		
	(4.90)	(5.32)			(0.19)	(0.71)		
Govexp^2^	−0.051 ***	−0.055 ***			−0.007	−0.013		
	(−5.37)	(−5.77)			(−0.95)	(−1.61)		
Govsize			4.608	6.971 **			0.912	0.874
			(1.36)	(1.97)			(1.44)	(1.62)
Govsize^2^			−1.405	−2.084 **			−0.341 *	−0.345 *
			(−1.44)	(−2.03)			(−1.72)	(−1.89)
_cons	3.841 ***	1.569 *	0.769	−2.451	1.572 ***	0.115	0.889	−2.741 ***
	(19.41)	(1.65)	(0.26)	(−0.79)	(4.52)	(0.12)	(1.51)	(−2.81)
Control variables	No	Yes	No	Yes	No	Yes	No	Yes
Year	Yes	Yes	Yes	Yes	Yes	Yes	Yes	Yes
Country	Yes	Yes	Yes	Yes	Yes	Yes	Yes	Yes
N	537	537	537	537	1283	1183	1399	1254
R^2^	0.643	0.664	0.624	0.644	0.361	0.428	0.334	0.346

Note: ***, **, and * indicate significance at the 0.01, 0.05, and 0.1 levels, respectively; Values in parentheses indicate t-statistics.

**Table 7 ijerph-19-07328-t007:** Heterogeneity Test: Countries with Different Levels of R&D Expenditure.

Variable	Patent	Patent	Patent	Patent	Patent	Patent	Patent	Patent
	(1)	(2)	(3)	(4)	(5)	(6)	(7)	(8)
Panel A: H_R&D					Panel B: L_R&D			
Govexp	0.174 ***	0.187 ***			−0.169	−0.109		
	(3.36)	(3.66)			(−0.55)	(−0.46)		
Govexp^2^	−0.020 ***	−0.023 ***			0.010	0.005		
	(−4.55)	(−5.22)			(0.46)	(0.29)		
Govsize			0.856 *	1.205 **			0.548	0.758
			(1.76)	(2.52)			(0.44)	(0.54)
Govsize^2^			−0.304 **	−0.459 ***			−0.207	−0.164
			(−1.98)	(−3.00)			(−0.49)	(−0.31)
_cons	2.168 ***	1.547 ***	1.818 ***	0.250	2.535 **	−0.545	1.549 *	−2.226
	(13.66)	(3.90)	(4.46)	(0.45)	(2.25)	(−0.16)	(1.91)	(−0.74)
Control variables	No	Yes	No	Yes	No	Yes	No	Yes
Year	Yes	Yes	Yes	Yes	Yes	Yes	Yes	Yes
Country	Yes	Yes	Yes	Yes	Yes	Yes	Yes	Yes
N	1396	1328	1500	1394	424	392	436	397
R^2^	0.457	0.506	0.405	0.472	0.345	0.441	0.340	0.437

Note: ***, **, and * indicate significance at the 0.01, 0.05, and 0.1 levels, respectively; Values in parentheses indicate t-statistics.

**Table 8 ijerph-19-07328-t008:** Robustness Test: Changing the Explained Variable.

Variable	Patent1	Patent1	Patent1	Patent1	Patent2	Patent2	Patent2	Patent2
	(1)	(2)	(3)	(4)	(5)	(6)	(7)	(8)
Govexp	0.182 ***	0.184 ***			0.120 **	0.153 ***		
	(3.01)	(3.05)			(2.34)	(2.97)		
Govexp^2^	−0.021 ***	−0.023 ***			−0.017 ***	−0.022 ***		
	(−4.11)	(−4.42)			(−3.99)	(−5.03)		
Govsize			0.773	1.013 *			0.949 *	1.267 ***
			(1.35)	(1.79)			(1.91)	(2.58)
Govsize^2^			−0.314 *	−0.398 **			−0.372 **	−0.543 ***
			(−1.73)	(−2.17)			(−2.39)	(−3.50)
GDP		−0.000 ***		−0.000 ***		−0.000		0.000
		(−4.24)		(−3.10)		(−0.71)		(0.64)
Pop		0.008 ***		0.007 ***		0.007 ***		0.006 ***
		(6.26)		(5.28)		(6.56)		(5.32)
HCI		0.715 ***		0.946 ***		0.387 ***		0.565 ***
		(4.39)		(5.68)		(2.78)		(3.91)
Xr		−0.000		−0.000		−0.000		−0.000
		(−0.56)		(−0.18)		(−0.31)		(−0.04)
CPI		−0.299		−0.590 ***		0.058		−0.145
		(−1.47)		(−2.91)		(0.47)		(−1.13)
Cp		0.066		0.038		0.023		−0.024
		(1.04)		(0.57)		(0.42)		(−0.41)
Sp		0.266		0.368 **		−0.033		0.118
		(1.48)		(2.00)		(−0.26)		(0.90)
_cons	2.120 ***	0.022	2.010 ***	−0.707	1.953 ***	0.686 *	1.486 ***	−0.112
	(11.13)	(0.05)	(4.23)	(−1.09)	(12.11)	(1.76)	(3.58)	(−0.20)
Year	Yes	Yes	Yes	Yes	Yes	Yes	Yes	Yes
Country	Yes	Yes	Yes	Yes	Yes	Yes	Yes	Yes
N	1513	1452	1594	1498	1742	1649	1847	1714
R^2^	0.243	0.289	0.217	0.280	0.484	0.531	0.445	0.507

Note: ***, **, and * indicate significance at the 0.01, 0.05, and 0.1 levels, respectively; Values in parentheses indicate t-statistics.

**Table 9 ijerph-19-07328-t009:** Robustness Test: Adding New Control Variables.

Variable	Patent	Patent	Patent	Patent	Patent	Patent
	(1)	(2)	(3)	(4)	(5)	(6)
Govexp	0.145 **	0.152 **	0.164 **			
	(2.30)	(2.38)	(2.58)			
Govexp^2^	−0.017 ***	−0.017 ***	−0.017 ***			
	(−3.17)	(-3.18)	(−3.29)			
Govsize				1.111 **	1.142 **	1.143 **
				(2.20)	(2.22)	(2.24)
Govsize^2^				−0.337 **	−0.331 *	−0.332 **
				(−2.01)	(−1.95)	(−1.97)
GDP	−0.000 ***	−0.000 ***	−0.000 ***	−0.000 **	−0.000 **	−0.000 ***
	(−3.09)	(−3.05)	(−4.15)	(−2.13)	(−2.16)	(−3.32)
Pop	0.009 ***	0.009 ***	0.004 **	0.008 ***	0.008 ***	0.003
	(7.90)	(7.82)	(1.99)	(6.67)	(6.63)	(1.41)
HCI	0.014	−0.004	−0.007	0.356 *	0.337	0.337
	(0.07)	(−0.02)	(−0.03)	(1.73)	(1.61)	(1.61)
Xr	0.000	0.000	0.000	0.000	0.000	0.000
	(1.10)	(1.15)	(0.37)	(1.33)	(1.35)	(0.59)
CPI	0.199	0.195	0.211	0.051	0.059	0.073
	(1.34)	(1.31)	(1.42)	(0.33)	(0.37)	(0.46)
Cp	0.044	0.053	0.046	0.015	0.021	0.015
	(0.78)	(0.93)	(0.82)	(0.26)	(0.34)	(0.24)
Sp	−0.111	−0.116	−0.124	−0.004	−0.016	−0.025
	(−0.68)	(−0.70)	(−0.75)	(−0.02)	(−0.09)	(−0.15)
Industry	0.005	0.005*	0.006 *	0.005	0.005	0.005
	(1.60)	(1.71)	(1.74)	(1.43)	(1.41)	(1.44)
Openess		−0.001	−0.001		−0.000	−0.000
		(−1.04)	(−1.27)		(−0.28)	(−0.46)
Emp			0.022 ***			0.022 ***
			(3.15)			(3.02)
_cons	1.211 **	1.258 **	0.954 *	−0.205	−0.230	−0.482
	(2.21)	(2.25)	(1.69)	(−0.31)	(−0.33)	(−0.70)
Year	Yes	Yes	Yes	Yes	Yes	Yes
Country	Yes	Yes	Yes	Yes	Yes	Yes
N	986	961	961	1035	1008	1008
R^2^	0.462	0.464	0.470	0.429	0.431	0.437

Note: ***, **, and * indicate significance at the 0.01, 0.05, and 0.1 levels, respectively; Values in parentheses indicate t-statistics.

**Table 10 ijerph-19-07328-t010:** Robustness Test: Poisson and Negative Binomial Tests.

Variable	Patent	Patent	Patent	Patent
	(1)	(2)	(3)	(4)
Model	Poisson	Poisson	Nbreg	Nbreg
Govexp	0.193 **		0.233 ***	
	(2.09)		(4.54)	
Govexp^2^	−0.015 *		−0.028 ***	
	(−1.71)		(−5.93)	
Govsize		6.089 **		4.378 **
		(2.39)		(2.19)
Govsize^2^		−1.754 **		−1.301 **
		(−2.37)		(−2.31)
GDP	−0.000 ***	−0.000 ***	−0.000 ***	−0.000 ***
	(−4.81)	(−5.60)	(−4.35)	(−4.04)
Pop	0.008 ***	0.008 ***	0.008 ***	0.008 ***
	(5.53)	(6.11)	(10.85)	(8.87)
HCI	0.758 ***	0.817 ***	0.564 ***	0.693 ***
	(3.66)	(4.04)	(3.42)	(3.97)
Xr	−0.000	−0.000	0.000	0.000
	(−1.01)	(−0.58)	(0.29)	(0.66)
CPI	0.310 *	0.357 **	−0.293 *	−0.565 ***
	(1.71)	(1.97)	(−1.87)	(−3.11)
Cp	−0.242	−0.283	0.069 ***	0.045
	(−0.47)	(−0.54)	(2.85)	(1.59)
Sp	0.134	0.216	−0.135	0.043
	(0.38)	(0.60)	(−1.03)	(0.32)
_cons	−3.426 ***	−8.582 ***	−2.857 ***	−6.378 ***
	(−11.05)	(−3.86)	(−12.04)	(−3.61)
Year	Yes	Yes	Yes	Yes
Country	Yes	Yes	Yes	Yes
N	1720	1791	1720	1791

Note: ***, **, and * indicate significance at the 0.01, 0.05, and 0.1 levels, respectively; Values in parentheses indicate t-statistics.

**Table 11 ijerph-19-07328-t011:** Robustness Test: GMM Estimators.

Variable	Patent	Patent	Patent	Patent
	(1)	(2)	(3)	(4)
L. Patent	0.628 ***	0.690 ***	0.555 ***	0.665 ***
	(28.62)	(45.32)	(17.98)	(74.49)
Govexp	0.287 ***	0.259 ***		
	(3.61)	(4.66)		
Govexp^2^	−0.029 ***	−0.018 ***		
	(−3.76)	(−2.87)		
Govsize			8.874 **	11.112 ***
			(2.32)	(9.80)
Govsize^2^			−2.167 **	−3.028 ***
			(−2.07)	(−9.77)
GDP	0.001	0.001	−0.001	0.001
	(0.51)	(0.52)	(−0.45)	(1.61)
Pop	0.002 *	0.008 ***	0.003 ***	0.009 ***
	(1.95)	(3.89)	(3.45)	(7.62)
HCI	0.294 ***	0.434 ***	0.761 ***	0.318 ***
	(2.62)	(7.42)	(4.60)	(3.62)
Xr	−0.001 ***	−0.001 ***	−0.001 ***	−0.001 ***
	(−3.42)	(−7.30)	(−5.75)	(-15.20)
CPI	0.465 *	-0.319 ***	0.608 *	−0.231 ***
	(1.85)	(-3.84)	(1.92)	(−4.80)
Cp	−0.659 **	−1.171 ***	−0.533	−0.547 ***
	(−2.10)	(−4.47)	(−1.49)	(−2.95)
Sp	0.182	1.580 ***	−0.127	0.679 ***
	(1.41)	(6.91)	(−0.81)	(3.77)
_cons	−0.213	−1.161 ***	−9.705 ***	−9.919 ***
	(−0.60)	(−6.14)	(−2.62)	(−11.11)
Year	Yes	Yes	Yes	Yes
Country	Yes	Yes	Yes	Yes
AR(1)-P	0.000	0.000	0.000	0.000
Sargan-P	0.697	1.000	0.987	1.000
Hasen-P	0.145	0.351	0.176	0.439
N	1720	1791	1720	1791

Note: ***, **, and * indicate significance at the 0.01, 0.05, and 0.1 levels, respectively; Values in parentheses indicate t-statistics.

**Table 12 ijerph-19-07328-t012:** Robustness Test: Driscoll and Kraay Standard Error Test.

Variable	Patent	Patent	Patent	Patent	Patent	Patent
	(1)	(2)	(3)	(4)	(5)	(6)
Govexp	0.086 **	0.111 ***	0.114 ***			
	(2.73)	(2.97)	(2.92)			
Govexp^2^	−0.013 ***	−0.016 ***	−0.016 ***			
	(−3.51)	(−3.70)	(−3.66)			
Govsize				0.939	1.216 *	1.224 *
				(1.60)	(1.93)	(1.98)
Govsize^2^				−0.345 *	−0.485 **	−0.490 **
				(−2.08)	(−2.67)	(−2.76)
GDP		−0.000	−0.000		−0.000	−0.000
		(−1.57)	(−1.67)		(−0.65)	(−0.65)
Pop		0.008 ***	0.008 ***		0.007 ***	0.007 ***
		(9.52)	(9.45)		(9.01)	(8.64)
HCI		0.396 ***	0.391 ***		0.700 ***	0.684 ***
		(3.69)	(3.59)		(4.87)	(4.70)
Xr		−0.000	−0.000		−0.000	−0.000
		(−0.37)	(−0.43)		(−0.13)	(−0.23)
CPI			−0.094			−0.279
			(−0.61)			(−1.66)
Cp			0.032			−0.006
			(1.50)			(−0.25)
Sp			0.107			0.205
			(0.70)			(1.44)
_cons	2.278 ***	0.992 ***	0.984 ***	1.700 ***	−0.306	−0.192
	(27.24)	(3.19)	(3.15)	(3.18)	(−0.44)	(−0.27)
Year	Yes	Yes	Yes	Yes	Yes	Yes
Country	Yes	Yes	Yes	Yes	Yes	Yes
N	1820	1726	1720	1936	1797	1791
R^2^	0.416	0.467	0.467	0.384	0.447	0.449

Note: ***, **, and * indicate significance at the 0.01, 0.05, and 0.1 levels, respectively; Values in parentheses indicate t-statistics.

## Data Availability

Publicly available datasets were analyzed in this study. This data can be found here: https://www.fraserinstitute.org/economic-freedom/dataset?geozone=world&page=dataset&min-year=2&max-year=0&filter=0; (accessed on 13 September 2021).

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
