# Peer review of "How Government Size Expansion Can Affect Green Innovation—An Empirical Analysis of Data on Cross-Country Green Patent Filings"

_ijerph, 2022, doi:10.3390/ijerph19127328_

Round 1

Reviewer 1 Report

The idea of the article is an interesting one and it is sustained by the methods of the work and the results for which the authors proved to be high-skilled in statistical software and interpretation. My only concern is that the model is too simple and if it cannot be improved right now by adding other extra variables, it should be mentioned in the conclusions part that this is something to be improved in future studies.

Reviewer 2 Report

1. The authors are encouraged to define the notion/process of „government expansion.” Several remarks on the consequences of „government expansion,” but the term remains unclear to the reader. Would you try a personal definition or confront the phenomenon with other scientific meanings to fix it?

2. The theoretical background must be correlated with methodology and study findings.  Even if the topic and the analytic design are pretty adequate, the paper looks more like a segmented structure without a fluid connection between theory, methodologic framework, and final remarks. 

3. Also, it is recommendable for the authors to provide a comprehensive explanation of the hypotheses and variables validation process. Policy recommendations are not fully connected to a theoretical foundation. The policy recommendations don't reflect the result of analytic operationalization in how they are highlighted. Consequently, an extensive intervention is needed here. 

Reviewer 3 Report

The aim of the study is to explore the impact that a government expansion has on a country level’s green innovation. The study also aims to understand how this works and whether this changes according to other factors (organisational inertia, R&D expenditure).

The topic of the study is interesting, where the study of policy-related green innovation needs clear directions and strong input from academics.

The purpose of the study is clear but needs to be better defined in the abstract. To make the abstract more attractive, it would be appropriate to introduce the topic in the first few lines.

The methodology section is well organised. To increase the clarity and quality of the paper, a figure representing the research design could be introduced.

The results are described in detail. A small description of how the results of the study will be dealt with could be introduced for a more immediate overview.

A critical discussion considering other studies in the literature is lacking. In my opinion I recommend implementing this aspect.

The conclusion section should be revised. The implications section should be implemented. From this detailed study, I expect more direction in terms of both policy and research.

The article is written in understandable English with concise and clear sentences.

Reviewer 4 Report

Dear authors,

I have had the pleasure of being invited to review your paper "How Government Size Expansion Can Affect Green Innovation? —An Empirical Analysis of Data on Cross-Country Green Patent Filings".

Attached you will find a set of recommendations/comments that might help you to revise your draft paper.

On a general note, you might wish to:

1. Clarify the key concepts that underpin your research

2. Distinguish between your contributions and the references to the existing body of literature (including providing the relevant sources/references that back your statements)

3. Spell out clearly the novel elements of your research and those findings that are in line with previous research

4. Properly explain the choice of variables and the assumptions your choices imply as well as the impact they have on your results

5.  Revisit the policy conclusions you derive from your analysis.

Thank you and best regards!

Round 2

Reviewer 3 Report

The revisions carried out improve the quality of the article.

Reviewer 4 Report

Dear authors,

Thank you for the responses to my previous set of comments.

I only have one additional question with regard to the following paragraph in page 2

Moderate expansion of government size could better compensate for market failures, 74 and guide resources such as funds and talents to be directed towards areas in urgent need 75 of development, especially in areas with high investment costs and high-risk factors such 76 as green technology innovation. For example, financial support and strengthening of leg- 77 islative protection could guide more resources to flow to the field of green technology 78 innovation. However, the excessive expansion of the government size, beyond the critical 79 value, and excessive intervention will inevitably damage social welfare and cause abnor- 80 mal economic development, which is similar to the planned economic system.

Is this part based on previous research? If so, please make sure you add relevant relevant references. Otherwise, is this paragraph part of your hypotheses? And does your research confirm them?

Thank you in advance and regards,

Anonymous reviewer
